# Local Treatment of Triple-Negative Breast Cancer: Is Mastectomy Superior to Breast-Conserving Surgery?

**DOI:** 10.3390/jpm13050865

**Published:** 2023-05-21

**Authors:** Alba Di Leone, Antonio Franco, Francesca Zotta, Lorenzo Scardina, Margherita Sicignano, Enrico Di Guglielmo, Virginia Castagnetta, Stefano Magno, Daniela Terribile, Alejandro Martin Sanchez, Gianluca Franceschini, Riccardo Masetti

**Affiliations:** Multidisciplinary Breast Center, Dipartimento Scienze della Salute della Donna e del Bambino e di Sanità Pubblica, Fondazione Policlinico Universitario Agostino Gemelli IRCCS, 00168 Rome, Italy

**Keywords:** triple-negative breast cancer, mastectomy, breast-conserving surgery

## Abstract

Triple-negative breast cancer (TNBC) is an aggressive type of breast cancer that lacks the expression of estrogen receptor (ER), progesterone receptor (PR) and human epidermal growth factor receptor 2 (HER2). TNBC accounts for about 15% of breast cancers and has a poorer prognosis as compared with other subtypes of breast cancer. The more rapid onset of this cancer and its aggressiveness have often convinced breast surgeons that mastectomy could provide better oncological results. However, there is no relevant clinical trial that has assessed differences between breast-conserving surgery (BCS) and mastectomy (M) in these patients. This population-based study aimed to investigate the distinct outcomes between conservative treatment and M in a case series of 289 patients with TNBC treated over a 9-year period. This monocentric study retrospectively evaluated patients with TNBC who underwent upfront surgery at Fondazione Policlinico Agostino Gemelli IRCCS, in Rome, between 1 January 2013 and 31 December 2021. First, the patients were divided in two groups according to the surgical treatment received: BCS vs. M. Then, the patients were stratified into four risk subclasses based on combined T and N pathological staging (T1N0, T1N+, T2-4N0 and T2-4N+). The primary endpoint of the study was to evaluate locoregional disease-free survival (LR-DFS), distant disease-free survival (DDFS) and overall survival (OS) in the different subclasses. We analyzed 289 patients that underwent either breast-conserving surgery (247/289, 85.5%) or mastectomy (42/289, 14.5%). After a median follow-up of 43.2 months (49.7, 22.2–74.3), 28 patients (9.6%) developed a locoregional recurrence, 27 patients (9.0%) showed systemic recurrence and 19 patients (6.5%) died. No significant differences due to type of surgical treatment were observed in the different risk subclasses in terms of locoregional disease-free survival, distant disease-free survival and overall survival. With the limits of a retrospective, single-center study, our data seem to indicate similar efficacy in terms of locoregional control, distant metastasis and overall survival with the use of upfront breast-conserving surgery as compared with radical surgery in the treatment of TNBC. Therefore, TNBC should not be considered to be a contraindication for breast conservation.

## 1. Introduction

Triple-negative breast cancer (TNBC) is a term that has been applied to cancers that lack expression of the estrogen receptor (ER), progesterone receptor (PR) and human epidermal growth factor receptor 2 (HER2). TNBC is a very heterogeneous disease. Lehman et al., in 2011, divided TNBC into six different subtypes: basal-like 1 (BL1); basal-like 2 (BL2); mesenchymal (M); immunomodulatory (IM); mesenchymal stem-like (MSL); and luminal androgen receptor (LAR) [1].

TNBC represents 15% of the 2,261,419 new cases of breast cancers diagnosed worldwide, which amounts to almost 300,000 cases each year [2]. Typically, TNBC exhibits rapid growth, higher aggressiveness, younger age at onset and worst rates of early locoregional recurrence (LRR) and distant metastasis (DM) [3]. A diagnosis of TNBC, as compared with estrogen receptor positive cancers, is more often clinical rather than through mammography [4]. Furthermore, these tumors are more frequently associated with hereditary breast and ovarian cancer syndromes, caused by loss of function germline mutations in one of two tumor-suppressor genes, BRCA 1 and BRCA 2 [1].

Breast-conserving surgery (BCS) including radiotherapy (RT) has been demonstrated in numerous clinical trials to provide at least equivalent prognosis to mastectomy (M) in breast cancer. These trials did not account for specific breast cancer subtypes such as triple-negative breast cancer (TNBC). Agarwal et al. conducted some long-term randomized clinical trials over several decades that found no statistically significant difference in long-term survival between breast-conserving surgery combined with postoperative radiotherapy and total mastectomy for early breast cancer patients [5].

There are few data in the literature on the possible impact of different surgical approaches on locoregional outcomes, distant metastasis and overall survival for specific breast cancer subtypes, including TNBC [6,7,8,9].

Most studies on TNBC have focused on the evolution of systemic treatments, while little attention has been paid to oncological outcomes in relation to locoregional treatment [9].

Breast cancer has led the way toward precision medicine increasing curation rates in patients with early disease and to prolong survival with an optimal quality of life in the metastatic setting. Very important advances have been achieved toward these goals due to the significant impact of immunotherapy on survival in triple-negative breast cancer and the exciting results of antibody drug conjugates.

The latest National Comprehensive Cancer Network (NCCN) guidelines express several recommendations on how to personalize drug treatment according to TNBC tumor subtypes, for example, poly-adenosine diphosphate-ribose polymerase (PARP) inhibitors are recommended in mutation carriers [8,10,11], while the addition of pembrolizumab and continuation of this agent after surgical treatment is recommended for patients with stage II or III TNBC receiving neoadjuvant chemotherapy. 

On the contrary, no specific indications are provided with regard to differentiation of surgical treatment in these patients, or tumor subtype-specific guidelines regarding adjuvant radiation therapy [7]. 

To address this lack of data, we conducted the present study to investigate the distinct outcomes in terms of locoregional disease-free survival (LR-DFS), distant disease-free survival (DDFS) and overall survival (OS) in patients with TNBC receiving upfront surgery either with breast-conserving surgery (BCS) or mastectomy (M).

## 2. Materials and Methods

### 2.1. Study Design 

We conducted a retrospective monocentric study that evaluated TNBC patients treated with upfront surgery at Fondazione Policlinico Agostino Gemelli IRCCS, in Rome. The observation period was from 1 January 2013 to 31 December 2021. 

We reviewed the clinical records of 305 consecutive patients with a histological diagnosis of primary TNBC, regardless of lymph node involvement and age or BRCA1 and BRCA 2 mutation carriers. 

Exclusion criteria included a history of cancer in the previous five years, metastatic disease and surgical treatment by lumpectomy not followed by adjuvant radiotherapy. We also excluded patients undergoing neoadjuvant chemotherapy with the aim of evaluating the effect of surgical treatment alone without the influence of systemic therapy. 

The patients were divided in two groups according to the surgical treatment undertaken: BCS (quadrantectomy and level II oncoplastic surgery) + adjuvant radiotherapy versus M (which included nipple and skin sparing M with immediate breast reconstruction as well as radical M without breast reconstruction), regardless of the clinical and pathological features of the neoplasm. The type of surgery was evaluated in a multidisciplinary meeting that included breast surgeons, plastic surgeons, radiologists, radiation oncologists, oncologists, and geneticists. Various factors influenced decision making regarding the type of surgery which included breast volume, ptosis, lesion size, multifocality, possible skin involvement, and the presence of pathogenic variants of BRCA 1 and BRCA 2 genes. With the introduction of level II oncoplastic surgery, even larger lesions may have benefited from breast-conserving surgery, while small lesions (cT1) in very small breasts or with multifocal lesions underwent radical surgery.

Then, the patients were stratified into 4 risk subclasses according to pathological staging: Group 1 included patients with breast cancer tumor size less than or equal to 20 mm without axillary lymph node involvement (pT1N0), Group 2 included patients with breast cancer tumor size less than or equal to 20 mm with axillary lymph nodes involvement (pT1N+), Group 3 included patients with breast cancer tumor size greater than 20 mm without axillary lymph nodes involvement (pT2-4N0), Group 4 included patients with breast cancer tumor size greater than 20 mm with axillary lymph nodes involvement (pT2-4N+).

The primary endpoint of the study was to evaluate LR-DFS, DDFS and OS in the various subclasses according to the type of surgical treatment.

### 2.2. Statistical Analysis

Continuous variables were described using mean ± standard deviation (SD) (median and interquartile range) and comparisons among the risk subclasses were performed using a Student’s *t*-test. Categorical variables were described by using absolute number and percentage and associations among them were assessed with the chi-square test. Survival curves were obtained using the Kaplan–Meier method and compared by using a log-rank test. Univariate and multivariable analyses were conducted using COX regression and were aimed at identifying predisposing factors to DDFS. All statistical evaluations were two-tailed and considered to be significant if *p*-value < 0.05 (*p* < 0.05). The statistical analysis was performed using the SPSS ver. 26.0 software (Statistical Package of Social Science).

## 3. Results

A total of 305 TNBC patients underwent upfront surgery at our center during the study period. Sixteen patients were excluded because of a previous diagnosis of malignancy. Among the remaining 289 patients, 247 patients (85.5%) underwent BCS (quadrantectomy in 227 cases and level II oncoplastic surgery in 20 cases) while 42 patients (14.5%) underwent M (conservative mastectomy in 25 cases and modified radical mastectomy in 17 cases) (Table 1).

Considering the final histological examination, significant differences were found concerning the size of the neoplasm, expressed as size (mm) (20.5 ± 10.5 vs. 31.3 ± 21.9, *p* < 0.0001); T staging (greater presence of pT3-4 in the M group than pT1 present more in the BCS, *p* < 0.0001); BRCA 1 or BRCA 2 pathological variants (*p* = 0.027); and histopathological grading (*p* = 0.019).

No significant differences were found regarding histotype (*p* = 0.277), presence of ductal carcinoma in situ (*p* = 0.833), axillary lymph nodes staging (pN, *p* = 0.568), mean age, menopausal status, multifocality/multicentricity, different histotype, presence of ductal carcinoma in situ and axillary lymph nodes staging (Table 2).

In order to overcome these biases, patients were stratified into four different risk subclasses on the basis of tumor size and axillary lymph nodes status (Table 3). No significant differences were found by dividing the patients into the four risk subclasses (T1N0, T1N+, T2-4N0, and T2-4N+ (*p* = 0.052)).

Table 4 underlines the univariate and multivariable analyses of the features predisposing to distant disease relapses. 

We found among the factors predisposing to distant relapses highlighted in the univariate analysis: pT1 (OR 0.303, *p =* 0.007), pT2 (OR 2.359, *p =* 0.036) and pN3 (OR 6.633, *p* = 0.010). Thus, the presence of a tumor less than or equal to 20 mm is protective for the risk of systemic disease recurrence. The type of surgery has no influence on the risk of distant disease recurrence.

Finally, in the multivariable analysis, we found two predictive factors of distant relapses: pT1 as favorable factors (OR 0.228, *p* = 0.034, 95% CI 0.058–0.898) and pN3 as unfavorable factors (OR 6.599, *p* = 0.012, 95% CI 1.511–28.818).

### Oncological Outcomes 

No significant differences regarding locoregional and systemic disease-free survival were observed between the BCS group and M group (Table 5). 

Similarly, no significant differences according to the type of surgery were seen in the the four risk subclasses with regard to LR-DFS (Group 1 (*p* = 0.333), Group 2 (*p* = 0.664), Group 3 (*p* = 0.542), Group 4 (*p* = 0.123)) (Figure 1).

Considering locoregional disease-free survival, breast-conserving surgery did not show significant differences with respect to radical surgical treatment in the four risk subclasses (Group 1 (*p* = 0.333), Group 2 (*p* = 0.664), Group 3 (*p* = 0.542), Group 4 (*p* = 0.123)) (Figure 1).

Considering distant disease-free survival, the two types of surgery showed no significant differences in every risk subclass (Group 1 (*p* = 0.457), Group 2 (*p* = 0.759), Group 3 (*p* = 0.590), Group 4 (*p* = 0.884)) (Figure 2).

Finally, considering overall survival, the two types of surgery showed no significant differences in every risk subclass (Group 1 (*p* = 0.682), Group 2 (*p* = 0.148), Group 3 (*p* = 0.705), Group 4 (*p* = 0.593)) (Figure 3).

## 4. Discussion

In the era of precision medicine, surgical treatment of breast cancer should be carefully planned and executed, taking specific characteristics into consideration, including tumor size and biology, size and shape of the breast and patient’s well-being. In fact, appropriate local treatment of breast cancer not only has a significant impact on the patient’s body, but also intensively influences the patient’s mental health and social interaction [12]. Breast-conserving surgery followed by radiotherapy is considered to be the gold standard for local treatment of invasive breast cancer [13].

When safely applied, breast-conserving surgery allows excellent cosmetic results with oncologic outcomes that overlap those achieved by M in terms of LR-DFS, DDFS and OS. In addition, breast-conserving surgery results in a lower rate of disability, particularly when utilized in older women [14]. However, to date, the randomized controlled trials that have compared outcomes between BCS and M have not selected patients according to their tumor subtype. With regard to TNBC, data available in the literature are very controversial. Several studies have indicated an increased risk of locoregional recurrence with the use of BCS, while others have not confirmed these data. Furthermore, studies have also documented that the incidence of locoregional recurrence in TNBC patients peaks by 36-48 months, but then decreases markedly [15,16].

Arvold et al. [17] reviewed 1434 patients who underwent breast-conserving therapy. In addition to age, they analyzed local recurrence according to breast cancer subtype, and they observed higher rates of local recurrence among HER2 and triple-negative subtypes, with a trend toward higher local recurrence among patients with luminal B subtype. They determined that 171 TNBC patients had a significantly higher risk of local recurrence as compared with patients with other subtypes (adjusted hazard ratio of 3.9, *p* = 0.001). 

Similar results were reported in a study by Nguyen et al. [9]: Among 793 patients who underwent BCT, 80 patients with triple-negative breast cancer had a consistently increased risk for local recurrence (adjusted hazard ratio of 7.1, *p* = 0.009). In addition, 5-year local recurrence was 7.1% in the TN group as compared with <2% for luminal A and B subtypes.

In addition, Zaky et al. [18] reported that, in their case series, TNBC patients showed significant increases in local and distant metastatic recurrence rates after BCS as compared with the other subtypes; in particular, patients with triple-negative tumors had higher local (12% versus 4% for non-triple negative) and distant recurrences (15% versus 4% for non-triple negative) rates (*p* = 0.01). 

On the contrary, a recent meta-analysis by Fancellu et al. that included 14 studies for a total number of 198,919 patients with TNBC, indicated statistically significantly lower odds of LRR among women who had BCS as compared with M. In the meta-analysis, lower odds of distant metastasis and a significantly lower hazard for all-cause mortality was also shown in women undergoing BCS versus M. They concluded that patients with TNBC who were selected for BCS did not show worse outcomes as compared with those treated with M, and that BCS could also be offered, when clinically feasible, in this category of patients [19]. 

Similarly, a meta-analysis by Wang et al. included a total of 15,312 breast cancer patients, of which 11,678 patients underwent BCT, and the other 3634 patients underwent mastectomy. However, only 4364 cases were identified as TNBC. The authors demonstrated that patients receiving BCS were less likely to develop LRR as compared with those receiving M, but this study did not compare patients by specific stage. The favorable outcome brought by breast-conserving therapy might be the contribution of the postoperative radiotherapy, which also contradicts the previous view that TNBC tumors exhibited high radio resistant [20].

Adkins et al., in a study of 1325 patients with TNBC that received either BCS or M, reported that the two different local treatments were at least comparable and the worst overall survival rate seen in these patients was due to the TNBC subtype itself and not to the type of local treatment received [21].

In a study by Gangi et al., breast-conserving therapy for TNBC was not associated with an increased risk of local recurrence as compared with non-TNBC subtypes. However, in their case series, the TNBC phenotype correlated with worse overall survival. They concluded that breast-conserving surgery also appeared to be appropriate for patients with TNBC [22].

In addition, Saifi et al. utilized a cohort of 12,761 patients and analyzed survival based on treatment modality using a propensity matched analysis (7237 patients had lumpectomy and radiotherapy, and 5524 patients had mastectomy only). They affirmed that radiotherapy, when added to lumpectomy, was associated with better OS as compared with mastectomy. A subgroup analysis showed that women younger than 40 had similar survival outcomes after mastectomy or breast conserving surgery and radiotherapy [23].

These discordant results leave a question open in the surgical decision-making process as to whether more demolitive local treatment can lead to better local and distant oncologic outcomes [24].

In our case series, we chose to examine only patients with triple-negative breast cancer that underwent upfront mastectomy or breast-conserving surgery (including quadrantectomy and level II oncoplastic surgery), so that the role of local treatment could be evaluated regardless of the influence of neoadjuvant chemotherapy. 

Therefore, we considered all patients undergoing upfront surgery from January 2013 to March 2021. We identified a total of 289 recruitable patients, of whom 42 patients (14.5%) underwent mastectomy and 247 patients (85.5%) underwent breast-conserving surgery. Among the latter, 20 patients (8.1%) benefited from oncoplastic surgery.

In our case series, the number of patients that underwent breast-conserving surgery was by far larger than the number of patients treated by mastectomy (85.5% vs. 14.5%). This is explained by an extensive use of oncoplastic surgical techniques, including level I and level II procedures, even when cancer excision volumes were larger, and the breast size allowed for major oncoplastic surgery reshaping.

In our group of patients that underwent breast-conserving surgery, 44.1% of the patients had a large tumor size (T2-T3-T4) but due to the use of oncoplastic surgery, mastectomy could be avoided.

We prefer to use BCS whenever feasible because it improves the quality of life of patients. In a recent review of a case series of 297 patients treated in our Institution either with BCS (87 patients) or M (210 patients) for large breast cancers of all subtypes, the BREAST-Q assessment showed statistically significant advantages in terms of chest pain, preserved breast skin sensitivity and physical well-being in a BCS and radiotherapy group as compared with a M group (*p* < 0.05) [25]. A multidisciplinary discussion, in a dedicated “surgery meeting” with a careful patient assessment and disease staging, is essential to select the best candidates for surgical treatment. The study underlined the concept that the surgical approach should be selected as a result of a shared decision-making process between a patient and physician, after thorough evaluation of long-term survival rates, risk of local recurrence, aesthetic issues and overall quality of life.

In our current study, when we compared outcome results based on surgical therapy alone, no difference was found between BCS and M (LR-DFS: 86.3% vs. 97.6% (*p* = 0.103), DDFS: 88.3% vs. 92.1% (*p* = 0.796) and OS: 69.2% vs. 93.3% (*p* = 0.652), respectively)

To overcome the bias that existed in the two groups in terms of tumor size (larger for patients undergoing M than BCS) (*p* < 0.0001), we re-evaluated our patients after stratification into four risk subclasses based on combined T and N pathological staging (Group 1 (T1N0), Group 2 (T1N+), Group 3 (T2-4N0) and Group 4 (T2-4N+)).

Considering locoregional disease-free survival, the two types of surgery showed no significant differences in every risk subclass (Group 1 (*p* = 0.333), Group 2 (*p* = 0.664), Group 3 (*p* = 0.542) and Group 4 (*p* = 0.123)). Similarly, no differences were observed in distant disease-free survival (Group 1 (*p* = 0.457), Group 2 (*p* = 0.759), Group 3 (*p* = 0.590) and Group 4 (*p* = 0.884)) and overall survival (Group 1 (*p* = 0.682), Group 2 (*p* = 0.148), Group 3 (*p* = 0.705) and Group 4 (*p* = 0.593)).

Radiation therapy certainly adds a significant contribution in obtaining similar results with the use of M or breast-conserving surgery. In general, radiation therapy is indicated for all patients with invasive carcinoma of the breast under the following conditions: received breast-conserving surgery, underwent mastectomy with breast cancer tumor size >5 cm or with positive margins, or underwent mastectomy with positive axillary nodes. Large population-based retrospective studies have shown that radiation therapy improved survival in triple-negative breast cancer patients, as well as in older women [26,27]. In fact, even in the population of women over 65 years old, in the TNBC tumor subtype, the benefit of radiotherapy treatment after surgery is undeniable.

In fact, as very precisely pointed out in a recent article by Zheng and al., the addition of radiotherapy to breast-conserving surgery ensures better local control as compared with mastectomy, since it typically encompasses the entire breast volume as well as the skin, subcutaneous lymphatic plexus, part of the pectoral muscle and local and regional lymphatics [28]. The authors also indicated that incidental irradiation may eliminate microscopic disease outside the breast field, reducing the risk of locoregional recurrence, and hence, the risk of distant relapse [29]. In fact, radiotherapy might eliminate possible involvement of microscopic lymph nodes that are not targeted by surgical treatment alone. 

In addition, the germline mutation of BRCA 1 and BRCA 2, which is common in patients with triple-negative breast cancer, appears to be able to inhibit tumor cells to repair DNA damage. This predisposition, which renders the tumor defective in DNA repair, has been argued to be a mechanism that could lead to increase radiosensitivity, which could explain the results associated with radiotherapy [30].

New evidence also suggests that radiation therapy could activate the immune system, producing immune modulatory effects through the induction of immunogenic cell death, which involves release of various cytokines and signals that modify the microenvironment of tumors and stimulate influx of immune cells to recognize tumor specific antigens released by dying cells, subsequently increasing the sensitivity of lymphocytes to tumor cells [30,31].

## 5. Conclusions

Although the retrospective and single-center nature of our study does not allow any definitive conclusion, our data seem to confirm a similar efficacy for breast-conserving surgery as compared with mastectomy in TNBC and that more extended local treatment does not produce survival benefits.

Further prospective randomized clinical trials should be conducted to optimize the treatment modality according to breast cancer biology.

Until conclusive data become available, surgeons should be swayed by higher aggressiveness of TNBC when selecting local treatment for their patients.

## Figures and Tables

**Figure 1 jpm-13-00865-f001:**
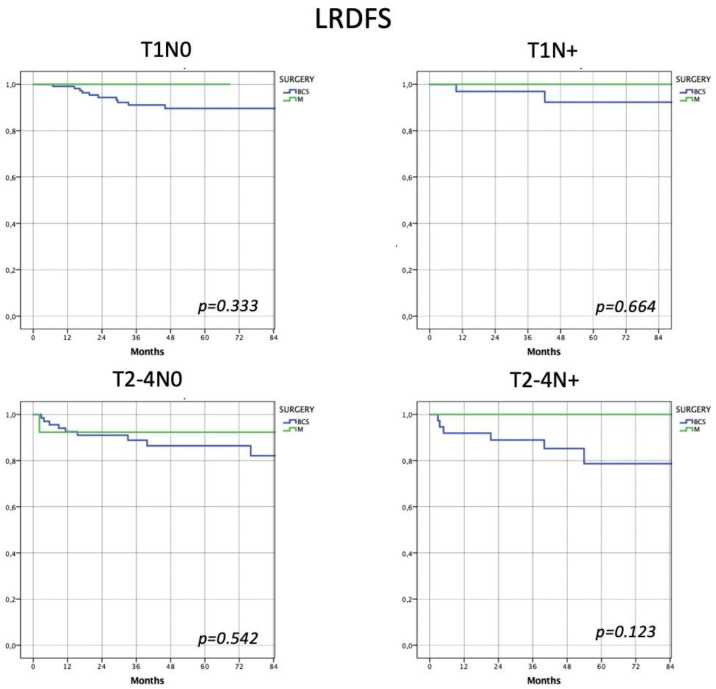
Assessment of locoregional disease-free survival according to the 4 risk subclasses. Patients who underwent the BCS or M showed no differences in terms of locoregional recurrence even when divided into the 4 risk subclasses.

**Figure 2 jpm-13-00865-f002:**
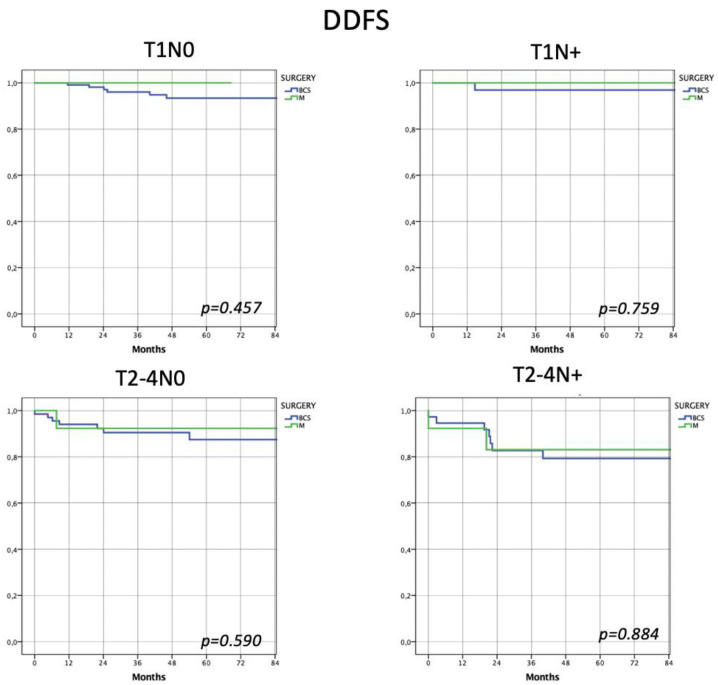
Assessment of distant disease-free survival according to 4 risk subclasses. Even considering distant disease recurrence, patients, regardless of risk class, show non-statistically different risk of recurrence.

**Figure 3 jpm-13-00865-f003:**
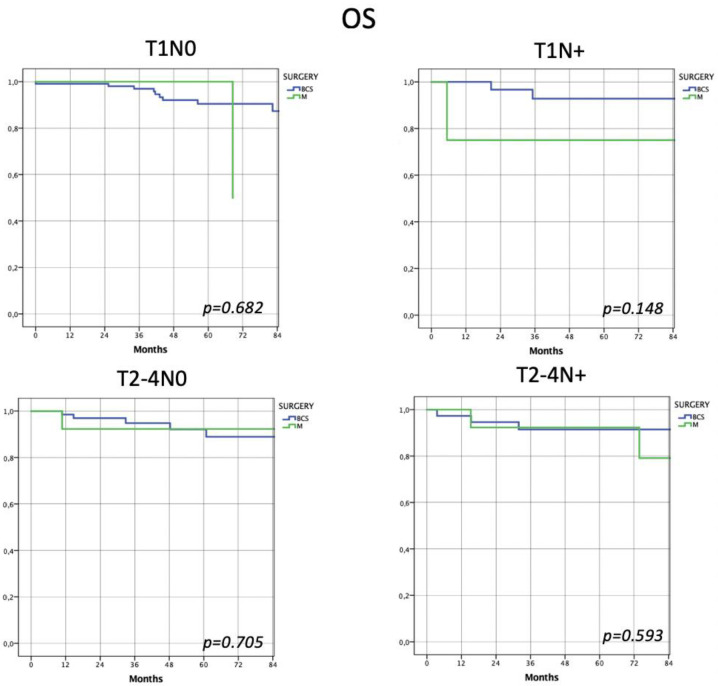
Assessment of overall survival according to 4 risk subclasses. Finally, patients undergoing the two types of surgery showed the same risk of overall survival, regardless of risk classes.

**Table 1 jpm-13-00865-t001:** Type of surgical treatment.

Conservative breast surgery	247 (85.5%)
- Quadrantectomy	227 (78.5%)
- Level II oncoplastic surgery	20 (6.9%)
Mastectomy	42 (14.5%)
- Conservative mastectomy	25 (8.7%)
- Modified radical mastectomy	17 (5.9%)

**Table 2 jpm-13-00865-t002:** Epidemiological, anatomical, biological and pathological features.

Characteristics	All 289 Patients	BCS 247 (85.5%)	M 42 (14.5%)	Significance
Mean age (years)	59.6 ± 14.2 (57.9–61.2)	59.6 ± 13.9 (57.9–61.4)	58.8 ± 15.8 (53.9–63.8)	*p* = 0.738
Menopausal status	207 (71.6%)	176 (71.3%)	31 (73.8%)	*p* = 0.854
BRCA 1/2 pathological mutations	31 (10.7%)	22 (8.9%)	9 (21.4%)	*p* = 0.027
Multifocality/multicentricity	38 (13.1%)	29 (11.7%)	9 (21.4%)	*p* = 0.134
Tumor size (mm)	20.8 ± 13.8 (19.1–22.4)	18.9 ± 10.3 (17.6–20.3)	32.4 ± 23.8 (24.4–40.3)	*p* < 0.0001
Istotype - DIC - LIC - IC NST - Not available	200 (69.2%) 2 (0.7%) 44 (15.2%) 43 (14.9%)	165 (66.8%) 2 (0.8%) 37 (15.0%) 43 (17.4%)	35 (83.3%) 0 (0%) 7 (16.7%) 0 (0%)	*p* = 0.006
Grading - G1 - G2- - G3 - Not available	10 (3.5%) 51 (17.6%) 182 (63.0%) 46 (15.9%)	9 (3.6%) 45 (18.2%) 148 (59.9%) 45 (18.2%)	1 (2.4%) 6 (14.2%) 34 (81.0%) 1 (2.4%)	*p* = 0.019
DCIS - Yes - No	53 (18.3%) 236 (81.7%)	46 (18.6%) 201 (81.4%)	7 (16.7%) 35 (83.3%)	*p* = 0.833
pT - pT1 - pT2 - pT3 - pT4	158 (54.7%) 14 (39.4%) 8 (2.8%) 9 (3.1%)	143 (57.9%) 96 (38.9%) 3 (1.2%) 5 (2.0%)	15 (35.7%) 18 (42.9%) 5 (11.9%) 4 (9.5%)	*p* < 0.0001
pN - 0 - 1 - 2 - 3	221 (78.6%) 45 (16.0%) 11 (3.9%) 4 (1.4%)	185 (77.4%) 41 (17.2%) 9 (3.8%) 4 (1.6%)	36 (85.7%) 4 (9.5%) 2 (4.8%) 0 (0%)	*p* = 0.568

BCS, breast-conserving surgery; M, mastectomy; DIC, ductal invasive carcinoma; LID, lobular invasive carcinoma; IC, invasive carcinoma; NST, no special type; DCIS, ductal carcinoma in situ.

**Table 3 jpm-13-00865-t003:** Stratification of patients according to T and N status.

	All 289 Patients	BCS 247 (85.5%)	M 42 (14.5%)	*p*-Value
Subclasses - T1N0 - T1N+ - T2-4N0 - T2-4N+	123 (42.6%) 36 (12.5%) 80 (27.7%) 50 (17.3%)	111 (44.9%) 32 (13.0%) 67 (27.1%) 37 (15.0%)	12 (28.5%) 4 (9.5%) 13 (31.0%) 13 (31.0%)	*p* = 0.052

**Table 4 jpm-13-00865-t004:** Univariate and multivariable analyses for distant relapses.

Characteristics	Univariate Analysis			Multivariable Analysis		
	OR	*p* Value	95% CI	OR	*p* Value	95% CI
Menopausal status	1.316	0.561	0.522–3.317			
BRCA pathological mutations	0.352	0.306	0.048–2.603			
Istotype	1.061	0.719	0.768–1.466			
Grading	1.367	0.300	0.757–2.468			
cT						
- 1	6.092	0.078	0.818–45.352			
- 2	0.238	0.238	0.032–1.759			
- 3	0838	0.672	0.370–1.897			
- 4	1.859	0.122	0.848–4.075			
cN	1.280	0.591	0.520–3.150			
Kind of surgery - CBS - M	1.172 0.853	0.796 0.796	0.351–3.918 0.255–2.851			
pT - 1 - 2 - 3 - 4	0.303 2.359 4.087 1.427	0.007 0.036 0.057 0.728	0.127–0.726 1.060–5.251 0.959–17.411 0.193–10.575	0.228 0.728	0.034 0.622	0.058–0.898 0.205–2.578
pN - 0 - 1 - 2 - 3	0.868 0.704 1.002 6.633	0.762 0.569 0.999 0.010	0.346–2.174 0.211–2.354 0.135–7.422 1.560–28.210	6.599	0.012	1.511–28.818

**Table 5 jpm-13-00865-t005:** Locoregional and systemic outcomes.

Outcomes	BCS (247–85.5%)	M (42–14.5%)	Significance
LR-DFS	27 (9.3%)	1 (0.3%)	*p* = 0.095
	86.3%	97.6%	LR = 0.103
DDFS	22 (7.6%)	4 (1.4%)	*p* = 0.778
	88.3%	92.1%	LR = 0.796
OS	17 (5.9%)	2 (0.7%)	*p* = 0.748
	69.2%	93.3%	LR = 0.652

LR-DFS = locoregional disease-free survival; DDFS = distant disease-free survival; OS = overall survival.

## Data Availability

Not applicable.

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
