# Peer review of "Local Treatment of Triple-Negative Breast Cancer: Is Mastectomy Superior to Breast-Conserving Surgery?"

_jpm, 2023, doi:10.3390/jpm13050865_

Round 1
Reviewer 1 Report
The authors rightly point out that this is a single centre study performed retrospectively. The results are not conclusive due to the nature of the study design. However, the results do confirm to an extent, conclusions of previous studies which support BCS over M. If the authors can discuss a prospective multicentric design and analysis plan which can support their conclusions, this should be included in the same.
Minor to moderate editing of language is definitely needed to improve the overall quality of the manuscript.
Author Response
Dear Editors,
We are pleased to receive your corrections that could make our paper entitled “The local treatment of triple negative breast cancer: is mastectomy superior to conservative surgery?” more suitable for publication in “Journal of Personalized Medicine”.
- In the following text, you will find our comments for each reviewer. (Our answers are reported according to the precise order of the questions raised by each reviewer)
- Reviewer #1:
Thank you for the suggestions of the Reviewer #1 to improve the quality of our paper. We appreciated your important contribution.
In this paper we have not described a prospective multi-center project but we will certainly plan one in the future.
English editing is done.
Sincerly
Reviewer 2 Report
As this is a retrospective study, it would be useful to include the information about systemic treatment regarding proportion of patients who received neoadjuvant chemotherapy. The response to neoadjuvant treatment may play a role in interpretation of their results.
Also, it is noted that almost 40% of patients with mastectomy were T1 tumors. Authors could comment on the rationale for selecting mastectomy over quadrantectomy for them.
Minor editing is recommended for grammar / printing errors, i.e. evidences instead of evidence etc.
Author Response
Dear Editors,
We are pleased to receive your corrections that could make our paper entitled “The local treatment of triple negative breast cancer: is mastectomy superior to conservative surgery?” more suitable for publication in “Journal of Personalized Medicine”.
- In the following text, you will find our comments for each reviewer. (Our answers are reported according to the precise order of the questions raised by each reviewer)
- Reviewer #2:
We appreciated the punctual observations of the Reviewer #2.
1) We did not mention neoadjuvant chemotherapy in the paper because we excluded this category of patients from our sample.
2) Patients affected by T1 tumors underwent mastectomy after surgical counseling because their own choice.
3) English editing is done.
Sincerely
Reviewer 3 Report
This study aimed to determine the impact of operation type on locoregional disease-free survival, distant disease-free survival, and overall survival in non-metastatic TNBC patients. They concluded that TNBC should not be considered a contraindication for breast conservation. The manuscript is generally well presented for the study design and writing. However, the novelty of this study is lacking and a statistical review is also needed before acceptance.
Other issues:
1. In the introduction section, the current understanding of the possible impact of different surgical approaches according to the literature review should be mentioned, and the current standard of care for local-regional treatment in TNBC as well.
2. Please double-check the format of all tables for data presentation in a more scientific manner.
3. Please explain the results in all figures more scientifically.
4. Please summarize and organize all the paragraphs into a more precise style, for example, put all the limitations of this study in one independent paragraph.
Author Response
Dear Editors,
We are pleased to receive your corrections that could make our paper entitled “The local treatment of triple negative breast cancer: is mastectomy superior to conservative surgery?” more suitable for publication in “Journal of Personalized Medicine”.
- In the following text, you will find our comments for each reviewer. (Our answers are reported according to the precise order of the questions raised by each reviewer)
- Reviewer #3:
Thank you for the suggestions of the Reviewer #3 to improve the quality of our paper.
- Statistical review was performed;
- Different surgical approaches according to the literature review was performed;
- All tables are corrected
- Results discussed more scientifically;
- All paragraphs are organized in more precise style.
Sincerely
Round 2
Reviewer 3 Report
I endorse the revised manuscript for publication.